



# Interdisciplinary Fracture Network Characterization in the Crystalline Basement: A case study from the Southern Odenwald, SW Germany

Matthis Frey[1], Claire Bossennec[1], Lukas Seib[1], Kristian Bär[1], Ingo Sass[1,2]

[1]Technical University of Darmstadt, Institute of Applied Geosciences, Department of Geothermal Science and Technology,
Schnittspahnstraße 9, 64287 Darmstadt, Germany.

[2]GFZ German Research Centre for Geosciences, Section 4.8: Geoenergy, Telegrafenberg, 14473 Potsdam, Germany

*Correspondence to: Matthis Frey (frey@geo.tu-darmstadt.de, ORCID: 0000-0003-2284-4215)*

**Abstract.** The crystalline basement is considered a ubiquitous and almost inexhaustible source of geothermal energy in the Upper Rhine Graben and other regions worldwide. The hydraulic properties of the basement, which are one of the key factors for the productivity of geothermal power plants, are primarily controlled by hydraulically active faults and fractures. While the most accurate in situ information about the general fracture network is obtained from image logs of deep boreholes, such data are generally sparse, costly and thus often not openly accessible. To circumvent this problem, an outcrop analogue study

with interdisciplinary geoscientific methods was conducted in the Tromm Granite, located in the southern Odenwald at the northeastern margin of the URG. Using LiDAR scanning, the key characteristics of the fracture network were extracted in a total of five outcrops, additionally complemented by lineament analysis of two different digital elevation models. Based on this, discrete fracture network (DFN) models were developed to calculate equivalent permeability tensors under assumed reservoir conditions. The influence of different parameters, such as fracture orientation, density, aperture and mineralization

was investigated. In addition, extensive gravity and radon measurements were carried out in the study area, allowing for more precise localization of fault zones with naturally increased porosity and permeability. Gravity anomalies served as input data for a stochastic density inversion, through which areas of increased open porosity were identified. A laterally heterogeneous fracture network characterizes the Tromm Granite, with the highest natural permeabilities expected at the pluton margin, due to the influence of large shear and fault zones.



## 1 Introduction

The Upper Rhine Graben (URG) represents region with a high potential for deep geothermal projects in Central Europe due to a significantly increased geothermal gradient of locally more than 10°C km$^{-1}$ (e.g. Agemar et al., 2014). Exploration, therefore, began already in the 1980s, allowing to build on decades of experience (Reinecker et al., 2019; Cuenot et al., 2008; Dezayes et al., 2005a). Convective heat transport along active large-scale fault zones has been identified as the main reason for the elevated temperatures at reservoir depths of 2 to 4 km (Bächler et al., 2003; Baillieux et al., 2013; Guillou-Frottier et al., 2013; Duwiquet et al., 2021). Besides, the thermal anomalies are enhanced by the radiogenic heat production, increased heat flux from the mantle and blanketing effect resulting from the low thermal conductivity of the thick sedimentary cover (Freymark et al., 2017). When exploiting the resulting vast potential, reservoirs with sufficient natural permeability are aimed at to ensure economical heat and power generation. In this context, the crystalline basement presents an attractive target due to the abundance of fractures and faults that enable substantial fluid flow (Glaas et al., 2021; Sausse and Genter, 2005a; Dezayes et al., 2021; Vidal et al., 2017). Examples of successful geothermal utilization of the basement for heat and power generation in the URG are the projects in Insheim, Landau, Rittershoffen and Soultz-sous-Forêts (Vidal and Genter, 2018). Fluid flow in fractured reservoirs depends on a multitude of parameters and processes, such as the density, orientation, length, opening and roughness of fractures, stress conditions or the influence of mineralization (Stober and Bucher, 2007; Bisdom et al., 2017; Ledésert et al., 2010; Meller and Ledésert, 2017). Consequently, it is essential to characterize and quantify the properties of the fracture network. The most reliable information, but unfortunately only as 1D dataset, is provided by deep wells of the above-mentioned geothermal projects, that penetrate the basement in the URG (Afshari et al., 2019; Baujard et al., 2017; Dezayes et al., 2010; Edel et al., 2018; Genter and Traineau, 1996; Glaas et al., 2021). However, these drillings are generally sparse and geophysical logs and cores are very costly and thus often not openly accessible. Additionally, fracture network information from wells does not contain information on the fracture length and depending on the borehole imaging method, the aperture and the kind of fracture mineralization can usually not be determined. In all areas with greater distance to deep boreholes with image-log data, the fracture network properties are therefore poorly constrained. To gain new insights and especially a better spatial and multiscalar understanding of the fracture network characteristics, the exposed crystalline rocks at the graben shoulders can be used as an outcrop analogue for the URG basement (Dezayes et al., 2021; Weinert et al., 2021). The presented study focuses on the Tromm Granite in the southern Odenwald (Fig. 1). This highly fractured granitic pluton is relatively homogenous with respect to lithology and representative for the predominantly granitoid basement in the northern URG (Frey et al., 2021a). The Tromm Granite is furthermore a promising site for the geothermal underground research laboratory (GeoLaB), where thermal-hydraulic-mechanical-chemical processes of deep geothermal reservoirs will be investigated in the future under in situ conditions to minimize the risk of future enhanced geothermal systems (Meller et al., 2018).

Characterization of fracture networks in the Tromm Granite is performed by following a multi-scale and interdisciplinary approach. Basement outcrops distributed over the entire pluton are analyzed using the LiDAR (Light Image Detection And



Ranging) technique (Biber et al., 2018; Fisher et al., 2014; Zeng et al., 2018). Visible fractures are identified and the relevant structural parameters are extracted. This dataset is supplemented by the examination of lineaments in two digital elevation

models (DEMs) with 1 m and 1 arcsecond resolution, resulting in a comprehensive description of the fracture network that extends over six orders of magnitude (e.g. Bertrand et al., 2015; Marrett et al., 1999; Pickering et al., 1995; Guerriero et al., 2011). Based on this dataset, discrete fracture network (DFN) models are developed, from which the hydraulic properties of the basement under reservoir conditions are inferred. Refined mapping of the potentially permeable fault and fracture zones is achieved by integrating surface gravity measurements (Deckert et al., 2017; Guglielmetti et al., 2013; Altwegg et al., 2015).

The stochastic inversion of these gravity data allow an accurate quantification of the porosity in the subsurface (e.g. Li and Oldenburg, 1998). Finally, radon measurements also indicate permeable fault zones (Ioannides et al., 2003; İnceöz et al., 2006; King et al., 1996; Jolie et al., 2015).

## 2 Geological framework

The crystalline Odenwald at the northeastern margin of the URG is the largest outcrop of the Mid-German Crystalline High

(MGCH), extending over 50 km from Heidelberg to Darmstadt (Fig. 1). This complex is usually subdivided into the 2 petrogenetic units Bergsträßer and Böllsteiner Odenwald, which are separated by a large-scale sinistral shear zone, called Otzberg Shear Zone (Amstutz et al., 1975; Stein, 2001). The older Böllsteiner Odenwald, located in the east, consists mainly of dome-shaped granitoid orthogneisses whose protoliths were emplaced during the Lower Devonian (Reischmann et al., 2001; Altenberger and Besch, 1993).

In comparison, the Bergsträßer Odenwald is dominated by Variscan plutonic rocks intruded into a metamorphic volcanic-sedimentary series (Altherr et al., 1999). The relics of these host rocks, commonly referred to as Schieferzüge (shale and gneiss bands), comprise gneisses, mica schists, amphibolites and scarcely marble (Okrusch et al., 1975). From north to south, the exposed basement rocks show a gradual transition from a primitive island arc regime to a collisional setting (Altherr et al., 1999; Okrusch et al., 1995). While the relatively older northern Frankenstein Complex (intrusion ages about 362 +- 7 Ma;

Todt et al., 1995) exhibits a primarily mafic composition, the southern plutons are predominantly felsic. For the latter, hornblende and biotite ages between 326 and 336 Ma and an intrusion depth of 15 to 19 km (0.4 to 0.5 GPa) were determined (Kreuzer and Harre, 1975). The emplacement of granitoids in the Carboniferous occurred during a syn-orogenic phase in an overall transtensional to extensional setting, as evidenced by large-scale strike-slip and normal faults, separating the individual magmatic and metamorphic units (Krohe, 1991; Krohe and Willner, 1995). These conditions are likely related to oblique

subduction of the Rheic or Rhenohercynian basins and associated back-arc spreading. Mineral alignment within the plutonic rocks indicates plastic deformation during crystallization (Krohe, 1992; Greiling and Verma, 2001). With progressive cooling, the increasingly brittle deformation was concentrated in large fault zones (Hess and Schmidt, 1989).





Figure 1: Overview of the study area: (a) geological map of the Odenwald (modified after HLUG, 2007); (a) geological map of the Tromm Granite in the southern Odenwald (modified after Klemm, 1900, 1928, 1929, 1933). Mapped fault zones have been compiled from various geological maps. HG = Heidelberg Granite, OZ = Otzberg Shear Zone, SA = Schollenagglomerat, SH = Sprendlinger Horst, TG = Tromm Granite, WP = Weschnitz Pluton.






The Tromm Granite forms an approximately 60 km² large wedge between the Weschnitz Pluton to the west and the metamorphic Böllsteiner Odenwald to the east. The Tromm Granite is a medium to coarse-grained, orthoclase-rich, biotite-bearing and often reddish granite, containing large potassium feldspar inclusions (Nickel, 1953; Maggetti, 1975). Locally, the rock gradually merges into granodiorite and mixed specimens can be observed. The southern part between Zotzenbach and Wald-Michelbach exhibits a fine-grained variety of the Tromm Granite of similar mineralogical composition and of younger

age than the coarser variety. In several locations, the granite is intruded by different generations of granitic, aplitic or pegmatitic dykes and veins (Klemm, 1933).

While an interlocking of the two plutons characterizes the contact between the Tromm Granite with the Weschnitz Granodiorite, the eastern boundary constitutes a 1-2 km wide heterogeneous westward dipping mylonitization and cataclasic zone along the Otzberg Fault (Schälicke, 1975; Hess and Schmidt, 1989). In the southeast, parts of this zone are overlain by

Buntsandstein. To the south, the pluton is bounded by the so-called Schollenagglomerat (Nickel, 1975, 1953). This is presumably a former Schieferzug, which was dismantled into separate blocks by shear movements along the Otzberg Shear Zone and/or by the intrusion of the Tromm Granite (Schälicke, 1975). The remaining amphibolite-facies overprinted rocks are often strongly intruded and assimilated by granitic dykes. The amphibolites are derived from mafic volcanic rocks or tuffs, whereas the gneisses and mica schists are rather of paragenic origin (Poller et al., 2001; Schubert et al., 2001; Todt et al.,

110   1995).

Most of the mapped and interpreted faults follow a N-S to NNE-SSW orientation. A secondary direction is present in the WNW-ESE. Lamprophyric dykes in the mylonites of the Otzberg Shear Zone, are dated at 330 Ma, suggesting that most of the shearing along this zone occurred shortly after the emplacement of the Tromm Granite (Hess and Schmidt, 1989). Later reactivations during Permian rifting or the opening of the URG in the Cenozoic are likely, as indicated by the vertical offset

of post-Variscan sediments at the eastern margin of the Tromm Granite, reaching locally several hundred meters (Klemm, 1933). It is generally difficult to assess the age of the faults where no sediments are preserved for correlation and mineralizations within the faults are not dated.

## 3 Material and Methods

The fracture network characterization is based on a multi-scale and interdisciplinary approach. In the following, the applied

methods are described. The first part focuses on structural geological investigations and DFN modelling. In the second part, the applied geophysical acquisition techniques are presented in detail. A summary of all investigations in the Tromm Granite is given in Fig. 2. In addition, the research data were deposited in the repository of the TU Darmstadt and are publicly accessible (Frey et al., 2021b).







**Figure 2: Overview map of the surveys conducted in the Tromm Granite: (a) locations of structural and geophysical data acquisitions; (b) detailed view of the combined gravity and radon profile; (c) detailed view of the quarry in Ober-Mengelbach with the location of 2D profiles, which have been manually interpreted. Digital elevation model provided by the HVBG (Hessische Verwaltung für Bodenmanagement und Geoinformation).**




## 3.1 Structural investigations

### 3.1.1 Lineament analysis

Two DEMs of the Tromm Granite were examined with respect to the density, length and orientation of lineaments. The high-resolution DEM with a cell size of 1 m allowed detailed structural investigations. In addition, the satellite-based SRTM model with a resolution of 1 arcsecond (c. 20 × 30 m) was used to identify regional structural features.

Lineaments are natural, rectilinear surface features that are uniquely identifiable and likely reflect subsurface structures, i.e.,

faults, discontinuities, or weakness zones. It should be noted that shallow dipping faults may not appear as linear structures and thus may be underrepresented, especially in areas of strong relief. However, most faults in the Tromm Granite are assumed to be rather steeply dipping.

The methodology of lineament analysis is described in previous studies, e.g. in Bertrand et al. (2015), Meixner et al. (2018) or Bossennec et al. (2021). The software QGis was used to generate hill shade maps of the DEMs, which were then visually

inspected for lineaments. To avoid misinterpretation, four different illumination azimuths of the hill shades maps (90°, 135°, 180°, and 225°) were compared and the results were checked for anthropogenic structures such as roads or other buildings. The digitized lineaments were used to calculate the lineament density P20 (number of fractures per unit area) and intensity P21 (total length of fracture per unit area) (Sanderson and Nixon, 2018).

### 3.1.2 Outcrop analysis

Five abandoned quarries located across the Tromm Granite were selected for detailed structural analysis of the fracture network (Fig. 2). As also described in Bossennec et al. (2021), the RIEGL VG 400 LiDAR instrument was used to generate high-resolution point clouds (point spacing ≤ 1 cm) of the outcrop walls. RGB information was captured with a Nikon camera and later mapped to the point cloud. Compared to classical scanlines, this approach allows for relatively quick acquisition of large structural datasets. At the same time, the statistical bias is reduced as all visible fractures are detected, not only those that cross

a 1D line. For an in-depth discussion of the reliability of LiDAR for outcrop analysis, see Biber et al. (2018), Fisher et al. (2014), Vazaios et al. (2017) or Zeng et al. (2018).

The raw LiDAR data were first imported into RiSCAN PRO to merge individual scans. Further analysis of the point clouds was performed using the open-source software CloudCompare. Vegetation was removed from the data both manually and by applying a noise filter. The point cloud was resampled to less than 2 million points to reduce the computational effort of the

following steps. Afterwards, the orientation of the surface normals was calculated by triangulation between the points and converted to dip and dip direction. Based on this, the RANSAC shape detection plugin was applied to automatically detect continuous fracture planes (e.g. Drews et al., 2018). The following parameters were chosen for this step: maximum distance to plane = 5 cm, scanning distance = 20 cm, maximum normal deviation = 10 °. Each detected plane was visually inspected and removed if it did not represent natural fractures.



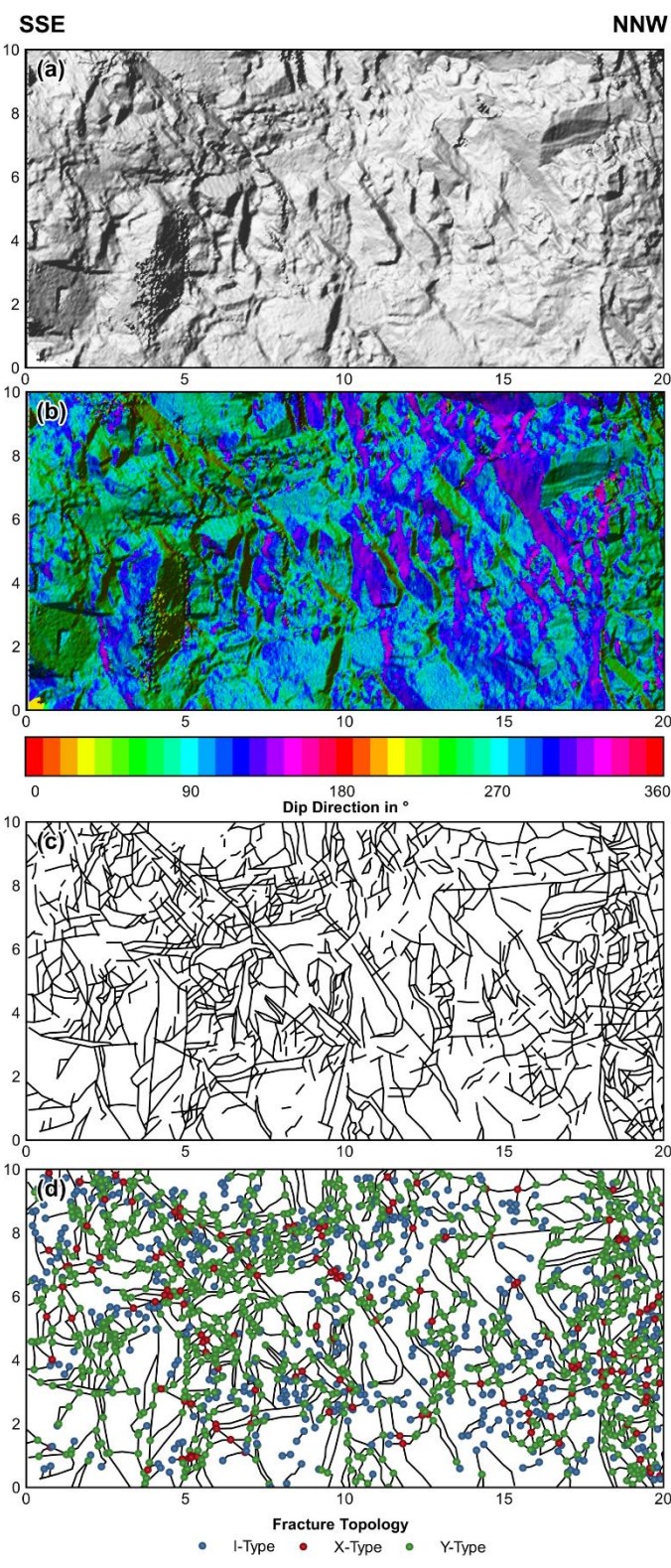





**Figure 3: Interpretation of a scanned outcrop wall from the quarry in Ober-Mengelbach (Profile A): (a) rasterized side view; (b) calculated dip direction; (c) manually interpreted fracture traces; (d) topology of the fracture nodes.**

Besides the automatic plane recognition, the LiDAR data were also manually interpreted in QGis (Fig. 3). For this purpose, side projections of the point clouds were rasterized and hill shade maps were again generated. Visible fractures were then

digitized to compute the fracture density P20 and intensity P21. Additionally, the linear fracture frequency P10 was extracted along virtual horizontal scanlines for each outcrop. The topology of the fractures was furthermore studied, to characterize the connectivity of the network. For this, the tips of all fracture branches were classified into 3 groups: isolated (I), abutting (Y) and crossing (X) nodes. The average number of connections per line $C_L$ was calculated from the number of nodes per type (Sanderson and Nixon, 2018).

The results of the lineament and outcrop analyses are finally summarized in a normalized trace length cumulative frequency plot with a power-law fitted to the data, which describes the relationship between frequency and the cumulative distribution of fractures lengths (Marrett et al., 1999; Ortega et al., 2006; Pickering et al., 1995).

### 3.1.3 DFN modelling

DFN models were generated with the software FracMan to quantitatively model the hydraulic properties of the fractured

crystalline basement, based on the structural parameters acquired in the field. Fracture orientations were implemented by performing a cluster analysis on the dip directions and dip angles extracted from the LiDAR data. The fracture density was defined along a virtual horizontal borehole using the calculated P10 values. The fracture length distribution was set according to the computed power law. A lower cut-off of 70 cm was applied, as significant censoring, i.e. under-representation of short fractures, occurs below this length (Fig. 6a). The effective fracture aperture largely governs the hydraulic conductivity of

fractures. Due to exhumation and weathering processes, measured aperture values at near-surface outcrops are usually not reliable (Place et al., 2016). Instead, an exponential distribution of the apertures was assumed, and three possible mean values (10 μm, 50 μm and 100 μm) were tested (Sausse and Genter, 2005b). A more accurate approach would be to relate the aperture to the normal stress on the fracture plane (Bisdom et al., 2017), but as the local stress magnitudes are largly unconstrained in the Tromm Granite, this was not pursued further. Apart from that, it has been shown repeatedly that a major part of the naturally

occurring fractures in the crystalline basement are mineralized at reservoir depth and therefore do not allow fluid flow (Genter and Traineau, 1996; McCaffrey et al., 1999; Dobson et al., 2003). For this reason, three different scenarios for the proportion of hydraulically active fractures in the DFN model were tested (1, 10 and 100%).

For a sufficient number of discontinuities, the fractured basement behaves like an anisotropic porous medium. The equivalent porous medium (EPM) permeability tensor can thus be calculated for a DFN model by e.g. the approach of Oda (1985). The

undisturbed rock matrix is considered impermeable (Jing and Stephansson, 2007; Weinert et al., 2020), implying that fluid flow occurs exclusively through connected fractures. The directional permeability is related to the size, orientation, opening and connectivity of the fractures. One key factor is the relationship between fluid flux along a fracture and the aperture, described by a cubic law (Snow, 1965). This relationship is based on the assumption of laminar flow between two parallel





surfaces, which is often not the case due to the irregular surface and aperture of the fractures and can therefore lead to errors.
The permeability tensors were computed along a regularly spaced grid with a cell size of 10 m to reduce the computational effort and afterwards, mean values were calculated for the entire DFN model. Different cell sizes between 1 m and 20 m were tested revealing no significant differences in the resulting mean permeability tensor.

## 3.2 Geophysical surveys

### 3.2.1 Gravity data/survey

During two surveys in summer 2020 and spring 2021, gravity measurements at 431 stations along 11 profiles have been conducted in the Tromm Granite (Fig. 2). Since a differential GPS was used to determine the position, the campaigns were restricted to the southern, less densely forested part. The GPS data were corrected against known fix points, resulting in about 10 to 20 cm positional accuracy. Gravity measurements were performed using the Scintrex® CG-6 Autograv gravimeter with an average station spacing of 100 m respectively 20 to 25 m close to presumed fault zones. Base measurements were taken
three times per day at a fixed station to record the instrument drift. A complete Bouguer anomaly was calculated for all gravity measurements by applying the standard correction density of 2.67 kg m$^{-3}$, which corresponds approximately to the mean rock density of the Tromm Granite (Weinert et al. 2021), as also confirmed by the Nettleton method. Particular focus was on the topographic correction, which along some profiles reaches up to 2 mGal due to the steep terrain. The calculation was performed with the software GSolve (McCubbine et al., 2018) in 3 separate zones (Zone 1: 0 to 2.1 km, DEM 10 m; Zone 2: 2.1 to 81
km, DEM 100 m; Zone 3: 81 to 167 km, DEM 1km). For the regional gravity signal analysis, about 5300 additional data points provided by the Leibniz Institute for Applied Geophysics (LIAG), Hessian Administration for Land Management and Geoinformation (HVGB), State Office for Geoinformation and Land Development Baden-Württemberg (LGL) and State Office for Surveying and Geographic Information Rhineland-Palatinate (LVermGeo) were used within a radius of 50 km around the survey area. Together with the newly acquired data, a Bouguer anomaly map with a nominal resolution of 20 m
was calculated using the minimum curvature interpolation method. A series of high-pass filters with cut-off wavelengths of 10 km, 5 km, and 2 km was then applied to subtract the regional gravity field.

### 3.2.2 Inversion of the gravity data

A stochastic 3D inversion of the high-pass filter Bouguer anomaly (10 km cut-off wavelength) was performed to infer the density distribution and the porosity in the subsurface. The commercial platform GeoModeller was used for this purpose,
which employes a Monte-Carlo Markov-Chain algorithm to invert geophysical data. A detailed discussion of the methodology is available in previous studies (Guillen et al., 2008; Frey et al., 2021a; Frey and Ebbing, 2020). The model domain has an extension of 7 km in E-W and 6 km in N-S direction and a depth of 2 km, approximately corresponding to the maximum depth that can be rsolved by the filtered gravity data. The upper boundary is defined by the high resolution 10 m DEM. A subdivision into several model units was not performed due to the relative homogeneity of the pluton and the lack of structural input data,



which is also the reason why the model lower boundary was set to a depth of 2 km. The continuous model was converted into a discrete cuboid voxel model with a cell size of 50 x 50 x 50 m. Further decreasing the cell size potentially resolves more details but exponentially increases the computational time of the inversion. As starting value for the rock density, a mean density of the Tromm Granite of $2.67 \pm 0.05$ g m$^{-3}$ was defined (Weinert et al., 2021).

The algorithm first calculates the geophysical effect of the starting model and then uses a Bayesian approach to determine the
likelihood of the model. In subsequent iterations, random variations of the model are generated according to the probability distribution of the rock density. Models that lead to a reduction in the misfit between calculated and measured gravity anomalies have a higher likelihood and are stored. After 250 million iterations, a larger collection of possible models is generated, allowing statistic evaluation.

Finally, the porosity can be calculated by rearranging the formula for the bulk density:

$$\rho_{bulk} = \rho_{fluid} \cdot \Phi + \rho_{matrix} \cdot (1 - \Phi) \tag{1}$$

$$\Phi = \frac{\rho_{bulk} - \rho_{matrix}}{\rho_{fluid} - \rho_{matrix}} \tag{2}$$

Where $\rho_{bulk}$ is the bulk density, $\rho_{fluid}$ the fluid density (approximately 1.0 g cm$^{-3}$), $\rho_{matrix}$ the matrix density and $\Phi$ the porosity.

### 3.2.3 Radon measurements

Radon is a naturally occurring radioactive gas that is concentrated in the soil air. The most abundant isotope with a proportion
of about 90 % is Rn-222 with a half-life of 3.82 days, formed in the decay series of U-238 (Baskaran, 2016). Because of this relatively fast decay, very deep sources in intact rocks are excluded. However, permeable fault zones may provide migration pathways where Rn-222 is transported from greater depth to the surface. Consequently, elevated radon concentrations are expected in the close vicinity of seismically or hydraulically active faults (Baskaran, 2016; Ioannides et al., 2003; Vazaios et al., 2017; Jolie et al., 2016).

Measurements of the activity concentration [Beq m$^{-3}$] of Rn-222 were carried out with the Saphymo AlphaGUARD 2000Pro at 20 points on one profile that crosses two presumed fault zones (Fig. 2b). Due to the comparatively high time-consumption, additional profiles were not investigated. Soil air was sampled using a hollow probe driven 1 m deep into the subsurface. A pump was connected to this probe, which flooded the ionization chamber of the radon monitor. The activity concentration was measured within 1-minute cycles. After 15 minutes, the air in the chamber was completely exchanged. The pump was then
switched off, and the chamber short-circuited for an additional 10 minutes. During this time interval, most of the very short-lived Thoron (Rn-220, $t_{1/2} = 55.6$ s) decayed so that the final reading corresponded only to Rn-222 concentration. Furthermore, soil samples were taken with a slotted probe at all stations to determine the soil type. Repeated measurements were performed at a base station to quantify the temporal variability of the concentration measurements.




## 4 Results

### 4.1 Fracture network properties

#### 4.1.1 Lineament distribution

Figure 4 and Table 1 compare the mapped faults in the Tromm Granite area with the lineaments extracted from the SRTM model and the DEM with 1 m resolution. A total of 30 faults with a characteristic length (arithmetic mean of the fracture length) of about 2900 m were mapped, corresponding to a P21 value of 0.0014 m m$^{-2}$. The total number of elements that were identified with the lineament analysis is significantly highly than the amount of mapped faults (177 for SRTM and 471 for the 1m DEM). Their characteristic length is consequently smaller with 1187 m and 680 m, respectively. The P21 is 0.0034 m m$^{-2}$ for the SRTM lineaments and 0.0051 m m$^{-2}$ for the lineaments of the high-resolution DEM. In all data sets, a spatially heterogeneous distribution of the faults or lineaments can be observed. The element density is highest in the eastern part of the Tromm Granite, i.e. in the area influenced by the Otzberg Shear Zone. The density is significantly lower in the west, especially for the mapped faults and the SRTM lineaments.

The main strike of the mapped faults ranges from N160°E to N170°E, which corresponds approximately to the direction of the maximum horizontal stress $s_{Hmax}$ determined at the eastern margin of the URG (Reiter et al. 2016). In contrast, the main set of SRTM lineaments strikes with N090°E±30°. The second subordinate set strikes with N030°E±10°. The strike directions of the lineaments from the high-resolution DEM show nearly an equal distribution. Only at N100°E a distinct cluster can be observed.

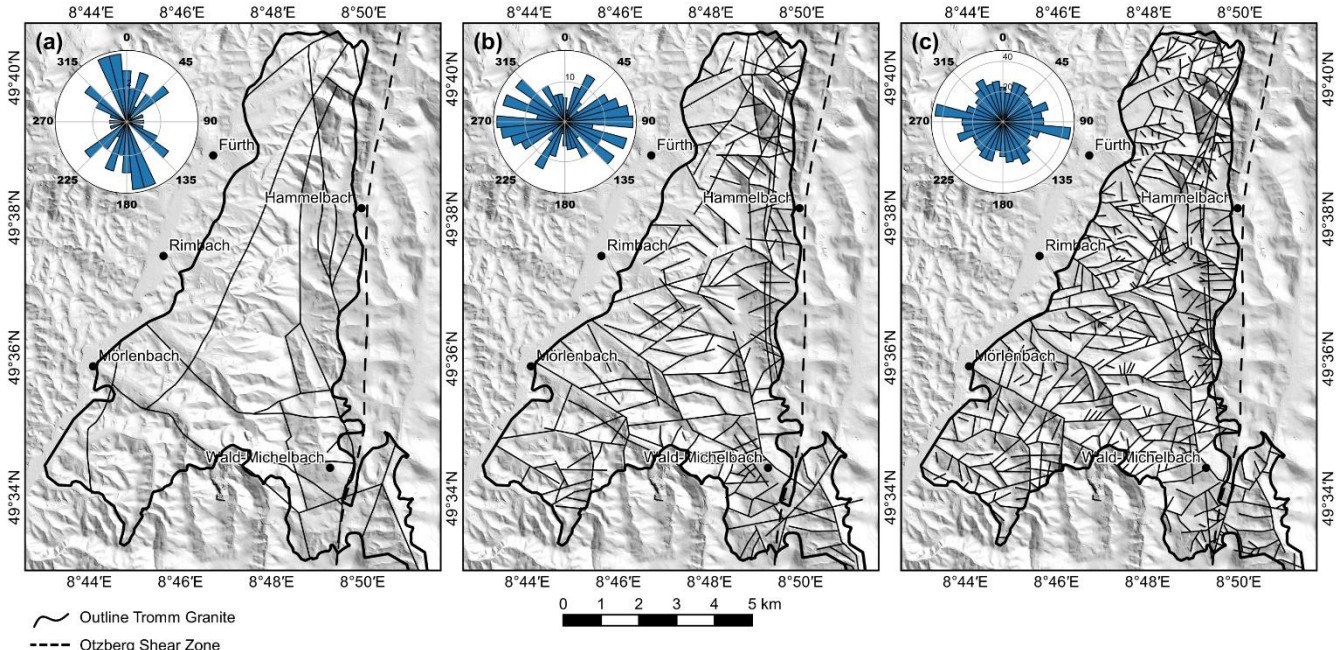



**Figure 4: Summary of the lineament analysis in the Tromm Granite area: (a) compilation of mapped faults from various geological maps (modified after HLUG, 2007; Klemm, 1900, 1928, 1929, 1933); (b) regional analysis using SRTM data (20 x 30 m resolution) (adapted from van Zyl, 2001); (c) local analysis using 1 m DEM (adapted from HVBG, 2021).**

### 4.1.2 Fracture networks in outcrops

In total five outcrops of varying size, distributed over the entire Tromm Granite area and hence representing the heterogeneity of the pluton, were investigated (Figs. 1 and 2). The two abandoned quarries Borstein and Streitsdölle in the central part of the study area are dominated by the typical, medium- to coarse-grained Tromm Granite. Remnants or intrusions of other rock species are scarce, but veins of younger granites can frequently be observed. These two outcrops have the lowest areal fracture intensity with a P21 of 2.43 m m$^{-2}$ and 2.83 m m$^{-2}$, respectively (Table 1). Conversely, the fracture characteristic length is the

longest here, reaching 1.28 m and 1.62 m respectively. The main set of fractures dips steeply and strikes about N160°E±20°, which corresponds to the main direction of the mapped faults in the Tromm Granite (Fig. 4). A second, subordinate set of steeply dipping conjugate fractures strikes N060°E±10°. Shallow dipping fractures are very rare.

  The most extended outcrop examined is an abandoned quarry of about 150 × 250 m close to the village of Ober-Mengelbach at the southern border of the Tromm Granite. In contrast to Borstein and Streitsdölle, the lithological conditions found here are

much more heterogeneous. There are meter- to ten-meter-wide amphibolite zones throughout the quarry that are highly deformed and intruded by granite or granodiorite. The magmatic contacts are usually not abrupt but rather characterized by mixed forms of amphibolite and granitoids. Again, several generations of granite intrusions occur. The distribution of fractures was investigated along four 2D profiles with a length between 20 and 30 m, and a height of 10 m (see Fig. 2 for location). The P21 ranges from 3.60 to 5.87 m m$^{-2}$ and is thus about twice as high as in the central Tromm Granite. The extraction of fracture

orientations using the Ransac filter was carried out for all outcrop walls to obtain the most comprehensive data set possible. Again, the primary set of fractures strikes N160°E±20° and a secondary set strikes about N070°E±10°.

  The two smaller outcrops in Hammelbach and the Weschnitz valley are located at the northeastern border of the Tromm Granite. Here, a fine- to medium-grained, cataclastic granite is predominant, which was considerably affected by the adjacent Otzberg Shear Zone. Consequently, the P21 is by far the highest with 10.82 m m$^{-2}$ and 9.07 m m$^{-2}$, respectively. The fracture

orientation also differs significantly from the other three locations. In Hammelbach, the fractures strike almost exclusively N100°E±20°. In the Weschnitz Valley at the northern margin of the Tromm Granite, two fracture sets were found, striking N050°E±10° and N130°E±20°, respectively. These directions correlate well with the orientation of the close-by mapped faults.





**Table 1: Summary of fracture network properties for all outcrop analyzed in the Tromm Granite area. OMB = Ober-Mengelbach.**

| Outcrop | Area [m²] | No. fractures | Min. length [m] | Max. length [m] | Char. length [m] | P10 [frac. m⁻¹] | P10 > 70 cm [frac. m⁻¹] | P20 [frac. m⁻²] | P21 [m m⁻²] | $c_L$ |
|---|---|---|---|---|---|---|---|---|---|---|
| **Faults & Lineaments** | | | | | | | | | | |
| Fault compilation | $62.7 \cdot 10^6$ | 30 | 195 | 9369 | 2899 | - | - | $4.79 \cdot 10^{-7}$ | 0.0014 | - |
| SRTM (1 arcsecond) | $62.7 \cdot 10^6$ | 177 | 238 | 4624 | 1187 | - | - | $2.84 \cdot 10^{-6}$ | 0.0034 | - |
| DEM 1m | $62.7 \cdot 10^6$ | 471 | 74 | 9001 | 680 | - | - | $7.53 \cdot 10^{-6}$ | 0.0051 | - |
| **Outcrops** | | | | | | | | | | |
| Borstein | 475 | 903 | 0.10 | 11.22 | 1.28 | 1.62 | 1.31 | 1.90 | 2.43 | 2.96 |
| Hammelbach | 67 | 1351 | 0.03 | 3.91 | 0.54 | 6.95 | 2.27 | 20.16 | 10.82 | 3.94 |
| Streitsdölle | 288 | 521 | 0.06 | 12.96 | 1.57 | 1.90 | 1.61 | 1.81 | 2.83 | 3.44 |
| Weschnitz Valley | 119 | 1332 | 0.03 | 7.57 | 0.81 | 7.05 | 4.25 | 11.19 | 9.07 | 4.09 |
| OMB total | 1050 | 5778 | 0.02 | 10.28 | 0.83 | 2.66 | 1.69 | 5.50 | 4.54 | 3.23 |
| OMB Profile A | 200 | 767 | 0.05 | 7.27 | 0.98 | 2.73 | 1.84 | 3.84 | 3.78 | 3.02 |
| OMB Profile B | 300 | 1647 | 0.06 | 6.24 | 0.89 | 3.17 | 2.07 | 5.49 | 4.89 | 3.11 |
| OMB Profile C | 250 | 2383 | 0.02 | 7.74 | 0.62 | 3.42 | 1.65 | 9.53 | 5.87 | 3.30 |
| OMB Profile D | 300 | 981 | 0.06 | 10.28 | 1.10 | 2.34 | 1.59 | 3.27 | 3.60 | 3.43 |



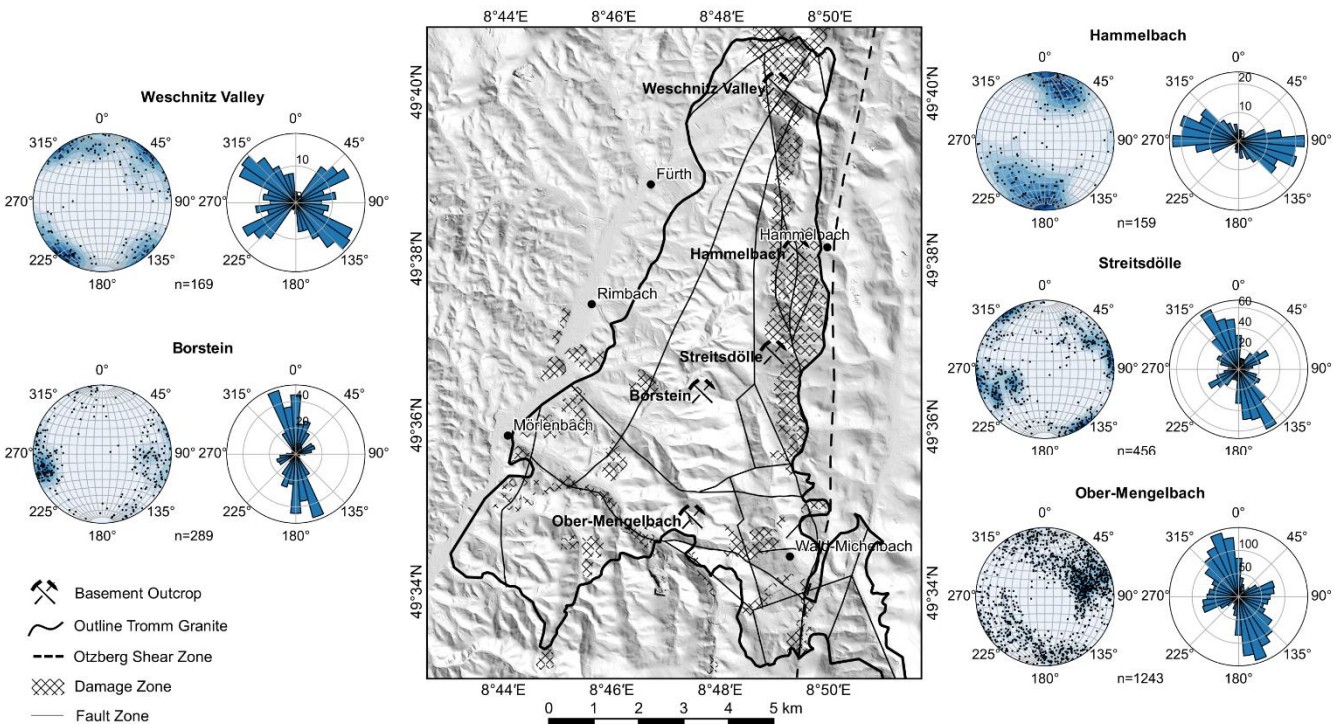


**Figure 5: Summary of the outcrop analysis in the Tromm Granite area (faults and outline of the Tromm Granite from HLUG (2007) and Klemm (1900, 1928, 1929, 1933)).**

### 4.1.3 Length distribution and fracture connectivity

Figure 6a provides a compilation of all identified fractures, mapped faults and lineaments from the Tromm Granite, with the
length plotted against the cumulative number of fractures normalized to the analyzed surface area. Length values range from
0.02 m to 9 km, hence covering about six orders of magnitude. The data points follow a power-law distribution with an
exponent of -1.96, which represents a typical value for fracture networks in the crystalline and especially granitic basement,
as shown in previous studies (Bertrand et al., 2015; Bossennec et al., 2021; Chabani et al., 2021). Deviations of the data from
this law can be explained by censoring and truncation effects. While not all small sized fractures can be identified due to the
limited resolution of the LiDAR point clouds, the full length of long fractures is often truncated at the edges of the visible
outcrop walls. To obtain the best power-law fit, a lower cut-off length of 70 cm for the outcrop fractures and 500 m for the
lineaments was therefore selected.

The IXY-topology was examined to quantify the connectivity of the encountered fracture networks, expressed by the average
number of connections per line $c_L$. An interpretation was done for the fractures on the outcrop scale and the lineaments on the
regional scale (Fig. 6b). All datasets plot in the lower-left corner of the IXY diagram, showing dominance of Y-nodes. The
proportion of I- and X-nodes ranges from 5 to 15 %, respectively. This node topology results in a $c_L$ of about 3 to 5, indicating
generally high connectivity of the fractures. Besides, a correlation between fracture intensity and connectivity can be seen.





The lowest $c_L$ (2.96) was determined for the Borstein outcrop, where P21 is also the smallest (2.43). In Hammelbach and in the Weschnitz Valley, the highest P21 (c. 9 - 11 m m$^{-2}$) and $c_L$ (c. 4) values were found. Another striking feature is the high

connectivity of the mapped faults, exhibiting almost no isolated ends, which might result from a bias of the mapping geologist.

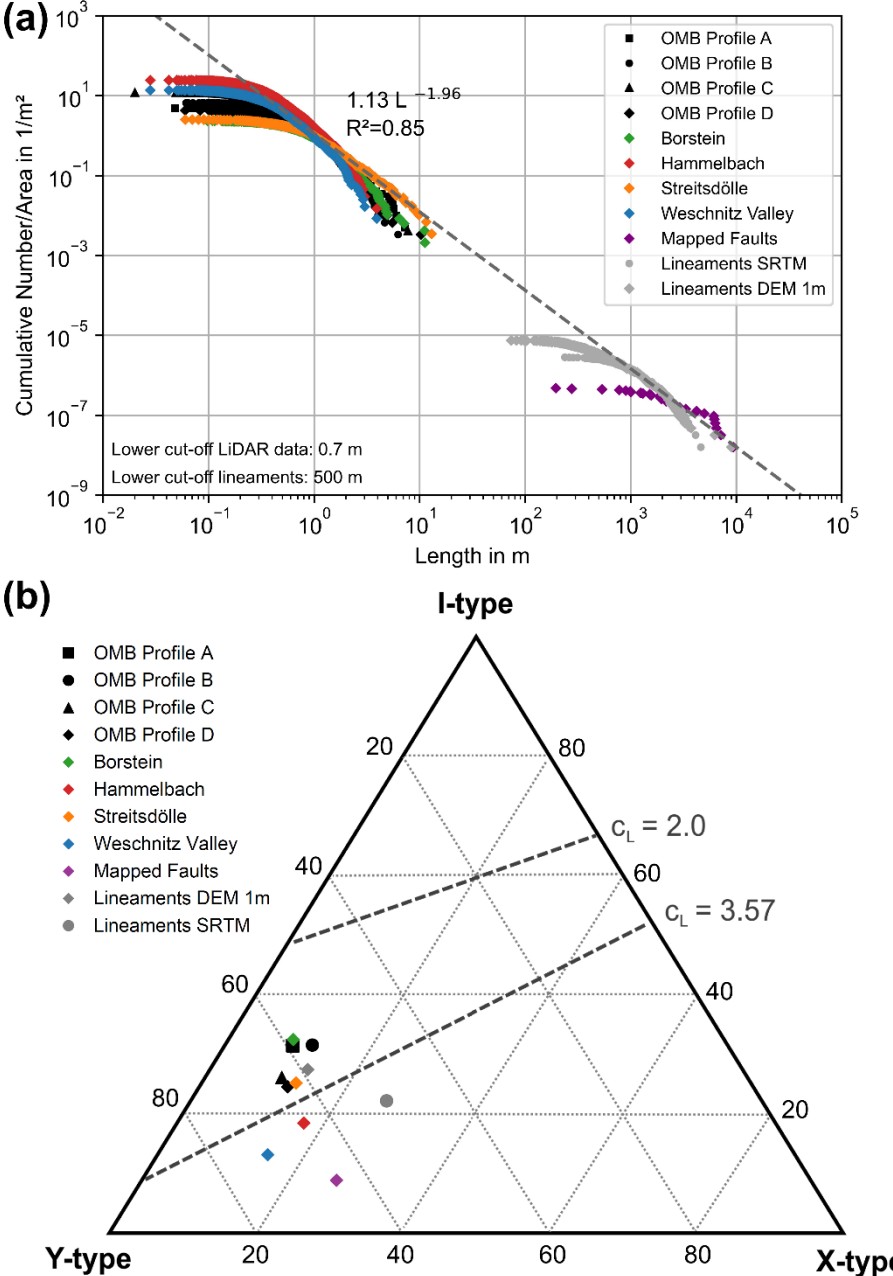

**Figure 6: (a) Area-normalized trace length cumulative frequency plot; (b) triangular plot of the proportion of I-, X- and Y-nodes of the analyzed outcrops.**



### 4.1.4 Results of DFN modelling

DFN models were created for the two outcrops Borstein and Weschntiz Valley, representing the end members of the Tromm Granite in terms of fracture density (Fig. 7). The calculated EPM permeabilities in x- (E-W-), y- (N-S-) and z-direction are summarized in Table 2. The fracture mean aperture has the largest influence on the permeability, as these two parameters are related via a cubic law. Practically speaking, this means that increasing the aperture by one order of magnitude leads to 3 orders of magnitude higher permeability. The proportion of open fracture, in contrast, is linearly related to permeability, i.e. a tenfold

increase also increases permeability by a factor of ten. The orientation of the fracture sets has furthermore a significant effect on the permeability of the basement. At Borstein, the permeability in the main direction of the fractures ($k_{yy}$) is almost one order of magnitude higher than perpendicular to it ($k_{xx}$). Finally, the difference between Borstein and Weschnitz Valley reaches again up to one order of magnitude, depending on the direction, for the same aperture and proportion of open fractures. This is mainly due to approximately four times higher fracture density in the second outcrop (Table 1).

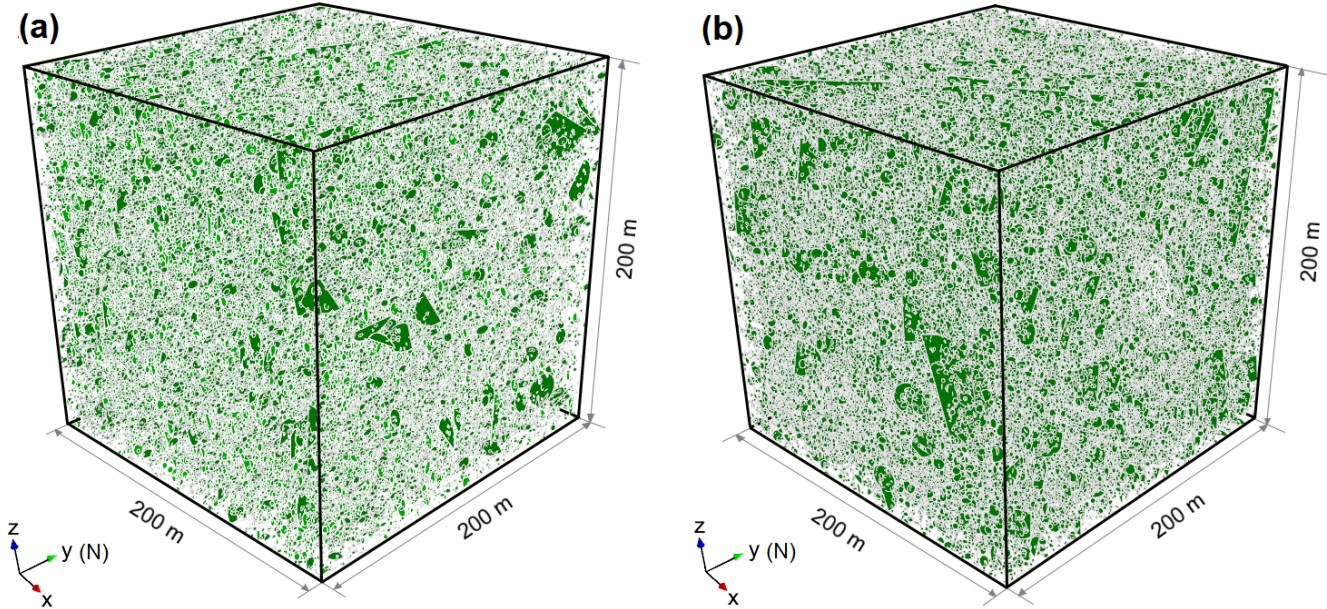


**Figure 7: Illustration of DFN models with 1 % open fractures for (a) the Borstein quarry (n = 214287) and (b) the Weschnitz Vally quarry (n = 279787).**

The question of which conditions represent deep geothermal reservoirs in 2 to 5 km depth remains to be answered. A comparison with data from Soultz-sous-Forêts and the United Downs project can adress this. Here, the mean permeabilities of

the granitic basement range from 1E-17 to 1E-15 m² (Egert et al., 2020; Mahmoodpour et al., 2021 in prep.). Accordingly, realistic permeabilities result for (1) an aperture of 10 μm when 10 to 100 % of the fractures are open and (2) for 50 μm at a maximum of 1 % open fractures. For 100 μm, the calculated permeabilities are too high in all cases and would thus rather represent permeabilities of single large-scale fractures or faults, while the smaller scale fractures would have smaller fracture apertures.



It should be noted that the hydraulic properties of fractured reservoirs are subject to strong spatial variations. For example, permeability can be increased by several orders of magnitudes close to active faults. In contrast, at larger distance from these faults or large-scale fractures, the mean permeability of the basement is rather in the order of 1E-18 to 1E-17 m².

**Table 2: Summary of the DFN modelling. The Oda permeabilities in x-, y- and -direction were calculated as a function of fracture density, orientation, aperture and proportion of open fractures.**

| | | Borstein | | | Weschnitz Valley | | |
|---|---|---|---|---|---|---|---|
| | | mean aperture | | | mean aperture | | |
| | m² | 10 μm | 50 μm | 100 μm | 10 μm | 50 μm | 100 μm |
| 100% open | $k_{xx}$ | 5.0E-16 | 6.3E-14 | 5.0E-13 | 3.8E-15 | 5.1E-13 | 4.1E-12 |
| 100% open | $k_{yy}$ | 2.1E-15 | 2.5E-13 | 1.9E-12 | 1.7E-15 | 2.2E-13 | 1.7E-12 |
| 100% open | $k_{zz}$ | 2.2E-15 | 2.6E-13 | 2.0E-12 | 4.5E-15 | 6.0E-13 | 4.7E-12 |
| 10 % open | $k_{xx}$ | 4.8E-17 | 6.2E-15 | 4.9E-14 | 4.0E-16 | 5.6E-14 | 3.4E-13 |
| 10 % open | $k_{yy}$ | 2.0E-16 | 2.5E-14 | 1.9E-13 | 1.7E-16 | 2.3E-14 | 1.5E-13 |
| 10 % open | $k_{zz}$ | 2.1E-16 | 2.6E-14 | 2.0E-13 | 4.7E-16 | 6.5E-14 | 4.0E-13 |
| 1 % open | $k_{xx}$ | 7.6E-18 | 1.9E-15 | 1.1E-14 | 5.2E-17 | 1.2E-14 | 5.7E-14 |
| 1 % open | $k_{yy}$ | 2.6E-17 | 3.2E-15 | 5.1E-14 | 2.6E-17 | 5.2E-15 | 2.5E-14 |
| 1 % open | $k_{zz}$ | 2.7E-17 | 4.4E-15 | 5.1E-14 | 6.6E-17 | 1.4E-14 | 6.7E-14 |

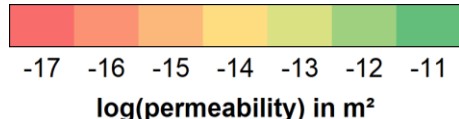

log(permeability) in m²


## 4.2 Gravity and Radon Anomalies

### 4.2.1 Bouguer anomalies

Figure 8 illustrates the results of the two gravity surveys. Within the Tromm Granite, Bouguer anomalies range from about - 10 to -5 mGal. The gravity field in this area is dominated by an NW-SE oriented trend (Fig. 8b) which was removed by
applying a series of high-pass filters with cut-off wavelengths between 2 and 10 km (Fig. 8c-e). The residual field exhibits distinct positive and negative anomalies. However, especially in the central part of the Tromm Granite, the lack of data points leads to considerable uncertainties.









**Figure 8: Results of the gravity survey: (a) Complete Bouguer anomaly map for the Tromm Granite (© Leibniz-Institut für Angewandte Geophysik, © Hessische Verwaltung für Bodenmanagement und Geoinformation); (b) low-pass filtered Bouguer anomaly with 10 km cut-off wavelength; (c) high-pass filtered Bouguer anomaly with 10 km cut-off; (d) high-pass filtered Bouguer anomaly with 5 km cut-off; (e) high-pass filtered Bouguer anomaly with 2 km cut-off.**

The strongest positive anomaly of 1 to 1.5 mGal is located north of Wald-Michelbach and coincides with a major lineament. Similarly, a positive anomaly of 0.5 to 1 mGal can be observed along the presumed fault zone between Zotzenbach and Wald-Michelbach. The strongest negative anomaly with an amplitude of about -0.5 to -1 mGal extends over several kilometres from SSW to NNE at the western boundary of the Tromm Granite to the Weschnitz Granodiorite. Another negative anomaly of about -0.4 mGal is located at the eastern boundary of the pluton, southeast of the village of Tromm. Here, the granite is highly fractured due to the proximity to the Otzberg Shear Zone, which is also indicated by the high concentration of local lineaments. Besides these larger anomalies, short-wavelength variations of the gravity signal in the range of -0.3 to 0.3 mGal occur on individual profiles, which is still significantly higher than the standard deviation of the gravity measurement and inversion result. A detailed comparison of the Bouguer anomalies with the measured Radon activity concentrations is given in Fig. 10.

### 4.2.2 Inversion results

Results of the stochastic gravity inversion are shown in Fig. 9. The inverted mean densities range from 2.52 to 2.84 g cm$^{-3}$ with an mean standard deviation of 0.0025 g cm$^{-3}$. At the top of the basement, the density distribution largely resembles the observed Bouguer anomalies. The most significant density decrease of 0.1 to 0.15 g cm$^{-3}$ is found at the western boundary of the Tromm Granite and corresponds to an open porosity of about 6 to 9%. In addition, smaller density decreases in the range of 0.03 to 0.1 g cm$^{-3}$ occur at the eastern boundary near the Otzberg Shear Zone, corresponding to an open porosity of about 2 to 6 %. Increased densities are mainly found in the area of the granodioritic Weschnitz Pluton and along the assumed fault in Gadern, north of Wald-Michelbach. Besides, very small-scale density variations are present in the south, which can be attributed to the lithological heterogeneity at the transition from Tromm Granite to the Schollenagglomerat Zone.

At greater depths, the inverted density model becomes more diffuse and the density variations are smaller. At 0 m a.s.l., the negative anomaly in the west and the positive anomaly in Gadern can still be clearly recognized. In contrast, the density reduction at the eastern edge is very weak. At 1000 m b.s.l., the variations have a very long wavelength and only range between -0.03 and 0.03 g g cm$^{-3}$. At 2000 m b.s.l., i.e. at the model base, the density is almost homogeneously distributed.



**Figure 9: Results of the gravity inversion. Difference between inverted and initial density at (a) the top of the basement, (b) 0 m a.s.l., (c) 1000 m b.s.l. and (d) 2000 m b.s.l.**





### 4.2.3 Comparison of gravity and radon measurements

A comparison of the radon activity concentration in soil air with the corresponding Bouguer anomalies is shown in Fig. 10

(see Fig. 2 for the location of the stations). A background activity of about 25 kBeq m$^{-3}$ was determined. The repeated base measurements furthermore revealed a standard deviation of 5 kBeq m$^{-3}$. Near the two assumed fault zones, a significant increase in the activity concentration can be observed with two pronounced peaks of 5 to 7 times the background value. The highest radon activity was measured in the zone between the two faults which coincides with a local negative Bouguer anomaly of about -0.1 mGal, indicating an increase in fracture intensity, respectively open porosity. The second peak is located close

to the northeastern fault zone, but here, as with the southwestern fault, a positive Bouguer anomaly is present.

The measured radon activity concentration depends strongly on the encountered soil type. In the valleys, e.g. in the area of the southwestern fault, water-saturated, clayey soils with low natural permeability are predominant, resulting in lower measured values. In contrast, higher radon concentrations tend to be measured in  loose, coarse-grained soils on the slopes.

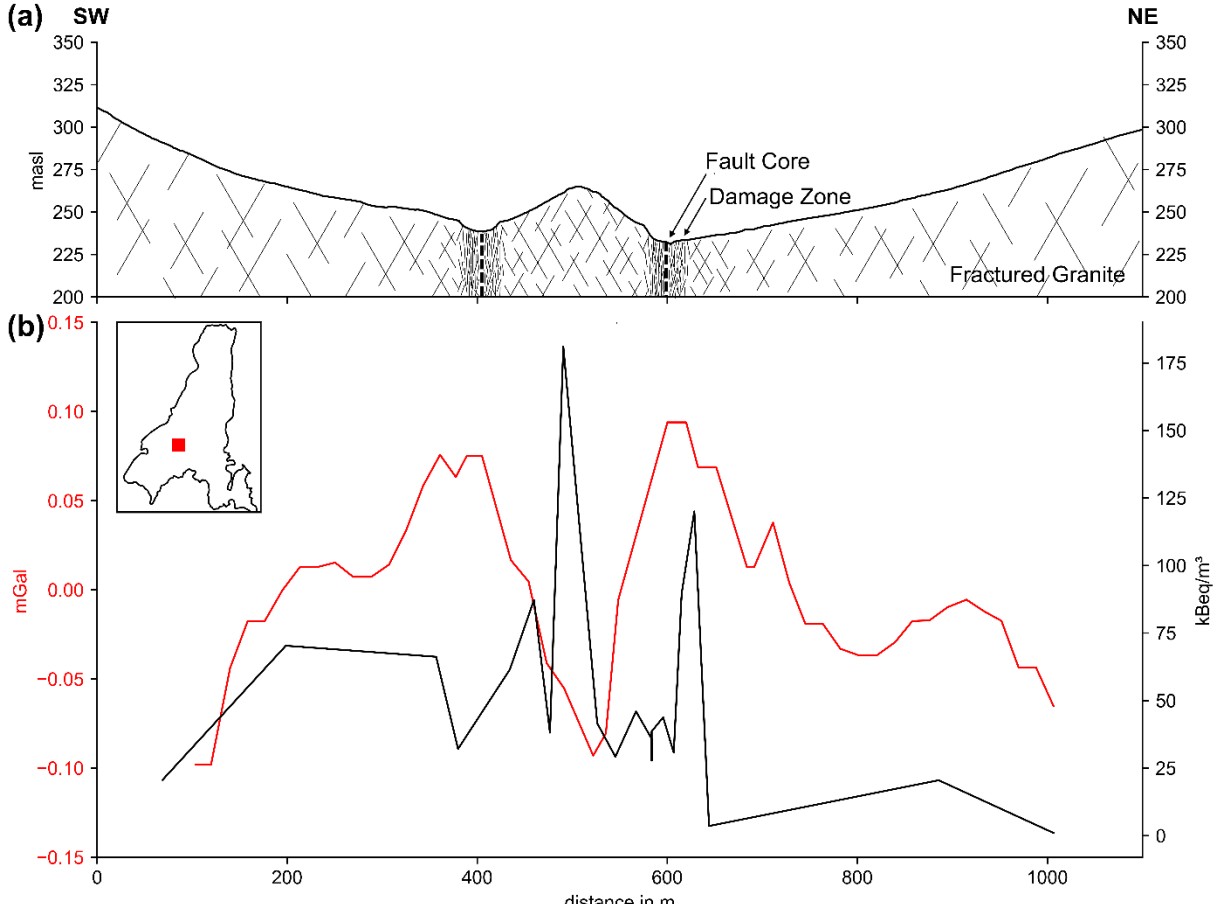

**Figure 10: (a) schematic cross-section of the investigated profile; (b) Comparison of the high pass filtered Bouguer Anomalies (red line) and measured radon concentration in the soil air (black line).The small map shows the location of the profile within the Tromm Granite (see also Fig. 2).**





## 5 Discussion

### 5.1 Fracture network characteristics

Based on the extensive structural geological investigations at the five outcrops and the lineament analysis, a comprehensive description of the fracture network in the Tromm Granite has been obtained. Scale independence of fracture length distribution was demonstrated with a power-law exponent of approximately -2. The length distribution is thus similar to other granitoid bodies in the URG region and other regions worldwide (Bertrand et al., 2015; Chabani et al., 2021). This multi-scale description could be extended by evaluating fractures on smaller rock samples or thin sections, but the applied methods already cover six

orders of magnitude.

Like the fracture length, the connectivity of the fracture network seems to be independent from scale or location. All outcrops and lineament maps indicate a dominance of Y-nodes, which is in clear contrast to the northern Odenwald, where I- and X-nodes represent the largest share (Bossennec et al. 2021). This can be attributed to different regional tectonic conditions during the intrusion, cooling and exhumation or to overprinting under variable stress conditions.

Compared to fracture length and connectivity, the orientation of the fracture sets shows some scale-dependent and spatial variations. In the outcrops Ober-Mengelbach, Borstein and Streitsdölle, the fracture orientations are controlled by the main fault direction of N160°E ±20°E in the Tromm Granite. Contrary to this, the fracture sets of the two outcrops Hammelbach and Weschnitz Valley are more influenced by local fault zones. Furthermore, the N-S trend of the mapped faults can hardly be found in the two lineament maps. Instead, elements perpendicular to it, i.e. oriented E-W, are dominant here.

Similar to the fracture orientation, the fracture density is subject to considerable lateral changes, which can be attributed to the influence of large-scale tectonic structures, especially at the pluton margins. In the eastern part of the Tromm Granite, the basement is deformed by the nearby Otzerg Shear Zone. As a result, there is an evident accumulation of lineaments and the outcrops show by far the highest fracture density. Medium fracture densities were found in Ober-Mengelbach, in the southern part of the pluton, i.e. at the border to the Schollenagglomerat. Although this area lacks pronounced long fault zones, the

lithological heterogeneities led to a more intense granite deformation than in the central Tromm Granite. Accordingly, the lowest fracture density was found in the outcrops Borstein and Streitsdölle.

In summary, the Tromm Granite is not characterized by a complete fractal fracture network. Although fracture length and connectivity seem to be scale-independent, orientation and fracture density show evident variations with scale and location. This fact must be considered when evaluating and modelling the basement, as the last two parameters can increase or decrease

permeability by up to one order of magnitude depending on the assumed mean apertures (Table 2).

### 5.2 Interpretation of gravity and radon anomalies

The measured gravity anomalies provide detailed insights into the density distribution in the subsurface of the Tromm Granite (Fig. 9). Negative anomalies of up to 1 mGal are concentrated at the western and eastern boundary of the pluton, where the basement is strongly deformed and fractured. Increased porosity of the granite, locally up to 9 %, is the most likely explanation





for the negative anomalies. Comparable results were obtained e.g. by Guglielmetti et al. (2013) for the Argentera Massif (NW
Italy). Alternatively, a several tens of meters thick layer of low-density Quaternary sediments or a thick basement weathering
horizon could lead to negative gravity anomalies. However, borehole data from the HLNUG suggest that Quaternary sediment
thicknesses typically do not exceed 10 to 15 m, accounting for a maximum of -0.2 mGal of the signal. Nevertheless, it is
possible that the weathering zone in the granite is locally 20 to 40 m thick, which could account for a gravity anomaly of up

to -0.5 mGal.

In the western part of the study area, the Tromm Granite is potentially structurally weakened at the contact with the Weschnitz
Pluton. Unfortunately, there are neither larger outcrops nor well data available, leaving this assumption speculative. The
slightly smaller negative anomaly at the eastern boundary to the Buntsandstein can be explained by the proximity to the Otzberg
Shear Zone. Here, the pluton is presumably characterized by similar structural properties as in the Hammelbach, and

Weschnitztal outcrops, which means the fracture density and thus the porosity are increased. Interestingly, the anomaly does
not extend over the entire damage zone at the eastern margin of the Tromm Granite but is concentrated in a limited area with
a high density of intersecting lineaments. The fracture porosity may be partially mineralized with e.g. barite (e.g. Tranter et
al., 2021), resulting in increased bulk density. Therefore, the gravity data allow a much better spatial resolution of the
potentially permeable fracture zones than the structural geological information alone.

Positive gravity anomalies of up to 1.5 mGal can be observed at the southern Tromm Granite along two fault zones. In Gadern,
several lamphropyric intrusions were mapped and, as in the quarry of Ober-Mengelbach, localized amphibolitic zones are
present. These mafic rocks have a considerably higher density than the Tromm Granite, which partly explains the gravity high.
Mineralization in fractures and faults e.g. by barite could also be responsible for the positive gravity anomalies. However, no
evidence of this was found in the field, only one barite vein was mapped by (Klemm, 1929) in Wald-Michelbach. Along the

fault between Zotzenbach and Wald-Michelbach, the positive gravity anomalies are mostly concentrated above the fault core
(Fig. 10), making mineralization the more likely explanation than the presence of mafic rocks. Quantifying fracture porosity
is challenging in both areas due to the heterogeneity of the basement. Such quantification requires a more accurate 3D
subsurface model as an input for the density inversion, which is currently not possible due to the lack of deep well data. An
aeromagnetic survey is plannend and may in the near-future provide valuable additional constraints on the structure and

composition of the Tromm Granite, which would improve the gravity inversion.

Radon measurements were carried out along just one profile due to high time-consumption of this method. Accordingly, a
regional interpretation of the results is only possible to a limited extent. Nevertheless, the determined radon anomalies provide
helpful insights into the architecture of the analyzed fault zones, especially in combination with the gravity data. Two distinct
radon peaks indicate localized permeable zones in the granite. The highest concentration correlates with a negative Bouguer

anomaly which further supports this assumption. Interestingly, the peaks are not located directly above the assumed position
of the faults, but in the damage zone a few metres to tens of metres to the sides. This observation suggests that the fault core
does not show increased permeability compared to the damage damage zone (Caine et al., 1996), e.g. due to mineralization,
which also explains the positive gravity anomalies.



### 5.3 Implication for deep geothermal exploration

The Tromm Granite in the southern Odenwald is, for several reasons, a well-suited outcrop analogue for the crystalline basement in the URG. On the one hand, the granitic body has a similar mineralogical composition as the reservoir rocks e.g. in Soultz-sous-Forêts or Rittershoffen (Dezayes et al., 2005b). Granitoids are dominant in the northern URG, as inferred from the joint inversion of gravity and magnetics data (Frey et al., 2021). On the other hand, the Tromm Granite was intruded in the same tectonic setting and overprinted under comparable conditions as the granites of the URG. Consequently, the orientation

and density of fractures in the deep boreholes can assumed to be similar to the Odenwald values.

Nevertheless, a direct transfer to deep geothermal reservoirs, where hydrothermal alteration and mineralization is to be expected, is challenging considering that weathering and exhumation significantly affect the near-surface fracture network. A DFN parameter study was carried out to estimate the hydraulic properties of the Tromm Granite under reservoir conditions (Table 2). Permeability of the basement is primarily determined by the aperture, in addition to the degree of mineralization

(considered here as proportion of open fractures), fracture density and fracture orientation. In the past, various approaches were developed to estimate the aperture as a function of fracture orientation and normal stress (Bisdom et al., 2017). However, this requires precise knowledge of the stress distribution, which is not available for the Tromm Granite due to the lack of deep boreholes or suitable outcrops (Heidbach et al., 2016). By considering the regional stress field, a rough estimate of the aperture might be possible but is beyond the scope of this study. In addition, DFN modelling suggests that only a small proportion of the total fracture network is contributing to flow. With an average aperture of 10 to 50 μm, probably only 1 to 10 % of the

fractures allow fluid flow, which fits well with obervations from Soultz sous Forets (Egert et al., 2020; Sausse et al., 2010). It should also be noted that $k_{xx}$ and $k_{yy}$ show a difference of up to one order of magnitude, which is particularly relevant for the planning of the well path trajectories of geothermal doublets. The deviated open-hole sections should preferably be oriented perpendicular to the fracture set with the highest permeability, thus intersecting as many of them as possible, allowing for high

flow rates.

Controlling induced seismicity during stimulation and operation represents a major challenge for deep geothermal exploitation of the crystalline basement (Rathnaweera et al., 2020; Zhang et al., 2013; Meller and Ledésert, 2017), as could be seen e.g. in Basel, Soultz-sous-Forêts, Landau, Insheim or Vendenheim. Stimulation is generally more feasible in areas with higher natural permeability, because lower injection pressures are required, resulting in a lower seismic risk.The highest permeability is

generally expected near large-scale fault zones with well-developed damage zones and hydrothermal overprinting. However, a large fault could respond to stimulation with a single, large-scale seismic event (Chang and Segall, 2016), as in Basel (Deichmann and Ernst, 2009), Pohang (Grigoli et al., 2018) or Vendenheim (Schmittbuhl et al., 2021), potentially bringing the geothermal project to an end. The structural geological investigations in Tromm Granite have shown that sufficient permeability may also occur in the outer damage zones of large faults, i.e. at a distance of several hundred meters to kilometers

from the fault core. In contrast to single long faults, the fractured host rock could respond with a higher number of low-magnitude events due to the smaller size of the reactivation areas, thus providing a lower overall seismic risk. This hypothesis



can be partially confirmed by data from Soultz-sous-Forêts (Dorbath et al., 2009). During stimulation of the well GPK2, a b-value of the Gutenberg-Richter law of 1.2 was observed, meaning that small and medium magnitudes were predominant. Moreover, the distribution of hypocenters did not reveal a clear structure, suggesting the reactivation of a dense 3D fracture network. In contrast, the simulation of the GPK3 wells produced fewer earthquakes, but with a higher average magnitude (b-value of 0.9). Furthermore, the distribution of the hypocenters showed defined structures that were identified as faults.

It was demonstrated that a combination of complementary methods can significantly reduce exploration uncertainties and helps to get a better understanding of a potential reservoir. Multidisciplinary reservoir characterization should also be considered more in the URG, where currently almost exclusively borehole and seismic data are used for prospection due to the higher resolution. High measurement density gravity and radon surveys should also be considered more to identify permeable zones in the subsurface, as they are relatively cheap to conduct in comparison to seismic surveys and easy to evaluate. Besides, magnetotelluric (MT) and controlled-source electromagnetic (CSEM) represent promising methods given the ability to resolve the convection of hydrothermal fluids in depth (Bär et al., 2020).

## 6 Conclusions

The applied multi-scale and interdisciplinary fracture network characterization of the Tromm Granite has led to the following conclusions:

- Combining outcrop and lineament analysis allows for a comprehensive description of the main fracture network characteristics.
- While fracture length distribution and connectivity are mostly scale-independent, fracture orientation and density vary significantly across the Tromm Granite. The latter two parameters are heavily affected by the crustal-scale Otzberg Shear Zone.
- Hydraulic properties of the fractured basement under reservoir conditions can be estimated with DFN models. However, the calculated permeabilities are associated with large uncertainties, as the stress conditions and, therefore, the fracture aperture are highly unconstrained.
- Gravity and radon measurements enable a more advanced mapping of potentially permeable zones. The fracture porosity can be inferred from the inverted density model, where homogeneous subsurface conditions are present. Lithological variations and mineralization prevent exact porosity quantification.
- Structural investigations and gravity anomalies show that the most suitable hydraulic properties are expected at the margin of the granitic pluton, where faults of regional extent influence the fracture network. The central Tromm Granite constitutes a relatively intact basement with significantly lower permeability, where permeability could only be created artificially by stimulation measures for geothermal utilization.



**Data Availability Statement**

The presented research data can be found at https://doi.org/10.48328/tudatalib-632 (Frey et al., 2021b).

**Contributions of each author**

Matthis Frey: Conceptualization, Data curation, Formal analysis, Investigation, Methodology, Validation, Visualization, Writing – original draft

Dr. Claire Bossennec: Conceptualization, Investigation, Methodology, Writing – review & editing

Lukas Seib: Investigation, Writing – review & editing

Dr. Kristian Bär: Funding acquisition, Project administration,Supervision, Writing – review & editing

Prof. Dr. Ingo Sass: Resources, Supervision, Writing – review & editing

**Competing Interest**

We have no conflicts of interest to disclose.

**Acknowledgments**

First of all, we would like to thank Cäcilia Boller for conducting one part of the gravity measurements. We thank Prof. Dr.
Eva Schill for the helpful discussions on the processing and interpretation of the acquired data. We are grateful that the HLNUG, LIAG, HVGB, LGL and LVermGeo provided the borehole, gravity and digital elevation data. We thank Sebastian Schröder und the municipalities of Wald-Michelbach and Rimbach for giving us access to the quarries in the Tromm Granite. We acknowledge support by the Deutsche Forschungsgemeinschaft (DFG – German Research Foundation) and the Open Access Publishing Fund of Technical University of Darmstadt.

**Funding**

This study was funded by the Interreg NWE Program through the Roll-out of Deep Geothermal Energy in North-West Europe (DGEROLLOUT) Project  www.nweurope.eu/DGE-Rollout). The Interreg NWE Program is part of the European Cohesion Policy and is financed by the European Regional Development Fund (ERDF).





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
