# Peer review of "Interdisciplinary Fracture Network Characterization in the Crystalline Basement: A case study from the Southern Odenwald, SW Germany"

_Solid Earth, 2021_

## Author Comment (AC1)

**Final Response: SE-2021-118**

**Anonymous Referee #2**

Thank you for reviewing our manuscript. In accordance with your comments, we have enhanced and extended in particular the discussion section. Please find below some more detailed responses to your comments.

*The study presents data on fracturing of intrusives considered as analogs of buried structures relevant for geothermal exploration. Technically speaking, the ms is well done and text well presented.*

*My overall concern is the lack of novelty of this work (and similar studies). There is no real innovation from a technical or methodological point of view (tools adopted are presently accessible to numerous groups) and the conclusions are so dependent on the assumptions and on parameters different from the subsurface that they are of little use.*

Answer: In the presented study we investigate the fracture network in a greenfield area. For this reason, it is important to use established methods in order to have a benchmark against previous studies. The novelty of the study consists mainly in the investigation of the specific outcrop analog, which has been poorly explored until now. In addition, such a combination of structural geological and geophysical methods has not previously been applied to characterize a potential fractured reservoir formation. The comprehensive data set is thus unique for the crystalline basement.

The data and interpretations are valuable for different geothermal applications in the crystalline basement, from EGS projects to medium-deep heat storages, as their efficiency largely depends on the architecture of the fracture network. In addition, the results are of high value to the GeoLaB underground research laboratory that is potentially to be constructed in the study area. To highlight this fact, an additional sparagraph has been included in the discussion.

We are aware that not all questions regarding the fracture network and the hydrogeological properties of the crystalline basement can be answered in the manuscript. Nevertheless, the study represents an important step forward and provides indications of where, for example, additional exploration measures are necessary.

Except for in Section 3.1.3 "DFN Modeling," no significant assumptions are made (and even these assumptions are based on published observations of other sites, see additional comment below)

*In addition, one could even argue that in those intrusives there are so many fractures that flow will take place anyway.*

Answer: While this may be true in the near-surface part of the pluton, the hydrogeological properties change considerably at depth (Manning und Ingebritsen 1999; Stober und Bucher 2007). On the one hand, this is due to the stress-induced closure of most fractures and, on the other hand, to the mineralization of the fractures. As a result, permeability at depths of some hundred meters can already be one or more orders of magnitude lower (Achtziger-Zupančič et al. 2017):

- Achtziger-Zupančič P, Loew S, Mariéthoz G. A new global database to improve predictions of permeability distribution in crystalline rocks at site scale. J. Geophys. Res. 2017;122(5):3513–39. doi:10.1002/2017JB014106.
- Manning CE, Ingebritsen SE. Permeability of the continental crust: Implications of geothermal data and metamorphic systems. Rev. Geophys. 1999;37(1):127–50. doi:10.1029/1998RG900002.
- Stober I, Bucher K. Hydraulic properties of the crystalline basement. Hydrogeol J. 2007;15(2):213–24. doi:10.1007/s10040-006-0094-4.

*- i have huge problems with the definition of sets in plots such as the ones in fig. 4: there are simply faults/fractures in most directions*

Answer: This fact is acknowledged in the manuscript: "The strike directions of the lineaments from the high-resolution DEM show nearly an equal distribution". The text was adjusted to make this clearer.

*- it is unclear to me what the role of gravity and radon studies; i did not see a link with the fractures. Even the gravity, radon and faults are less correlated than pretended*

Answer: The introduction and discussion of the radon and gravity measurements was revised to emphasize the link between geophysical and structural data. We added a more thorough description of the uncertainties associated with this approach.

*- i miss a sensitivity analysis and cannot therefore judge how robust are the conclusions*

Answer: In section 4.1.4 the effect of various parameters on the hydrogeological properties were systematically analyzed, as presented in the manuscript. A more detailed sensitivity analysis would mean a disproportionate additional effort, as the DFN is very time consuming and cannot be automatized. Hence, we added references to other sensitivity studies on this topic that are supporting our interpretations.

*- i also miss information on the outcrops: are they vertical? horizontal? in between?*

Answer: The information was added to the text.

*What to do? Either the authors can substantially highlight innovative and useful results, or it remains a technically interesting study with little impact.*

Answer: We are somewhat surprised that the results are considered to be not useful. Please see the first comment above where we emphasize the importance of the study. We have revised the introduction, discussion, and conclusions to enhance the impact of the manuscript.

Interpretation of data from outcrop analog studies with respect to deep reservoir properties are always subject to uncertainties. However, with very little well data available (see introduction), this approach is the only possibility for gaining new insight into the structural/hydrogeologic/geophysical properties of the basement without drilling cost-intensive new wells.

---

## Author Comment (AC2)

**Final Response: SE-2021-118**

**Anonymous Referee #1**
General Review

*The manuscript proposes an extensive structural study of the Tromm Granite with various approaches: lineament analysis, LiDAR implemented on outcrops, DFN model, gravity and radon measurements. The manuscript is written in understandable manner and identify potential targets for future geothermal projects in the Odenwald area. An effort should be done in the description of the methodology in order to clarify for which purpose the different approaches were chosen and how they contribute to the final interpretation.*

Answer: Thank you very much for the review of our manuscript. Your comments were very helpful to improve the text. We adjusted the text of section 3.1.2 to clarify the methodology.

*The main issue of this interdisciplinary study is the comparison of results from different scales that sometimes is not relevant. To be clear, using data at the regional scale or at the outcrop scale to interpret geothermal circulations in a deep reservoir is not trivial. For example, they work with radon measurements and Bouguer anomaly implemented on an outcrop to interpret permeability of fault zones. The interpretation of gravity data in term of permeability is very smooth and large scale. It is more adapted for permeability at regional scale than a reservoir one as suggested in the manuscript. Furthermore, interpretation of permeability of a fault zone observed on an outcrop can not be directly transferred to the reservoir.*

Answer: We agree that the direct transfer of our results to the crystalline basement in the URG under reservoir conditions is challenging. Indeed, this is the case for any analogous study and has been discussed in detail, e.g., by Alexander (1993), Homuth & Sass (2014), or Howel et al. (2014). Nevertheless, the study provides important insights into the architecture of the fracture network of the crystalline basement in the immediate vicinity of the Upper Rhine Graben and indicates where increased permeabilities are likely to occur.

- Alexander, J. (1993). A discussion on the use of analogues for reservoir geology. Geological Society, London, Special Publications, 69(1), 175-194.
- Homuth, S., & Sass, I. (2014, February). Outcrop Analogue vs. Reservoir Data: Characteristics and controlling factors of physical properties of the upper Jurassic geothermal carbonate reservoirs of the Molasse Basin, Germany. In Thirty-Eighth Workshop on Geothermal Reservoir Engineering (pp. 24-26).
- Howell, J. A., Martinius, A. W., & Good, T. R. (2014). The application of outcrop analogues in geological modelling: a review, present status and future outlook. Geological Society, London, Special Publications, 387(1), 1-25.

Regarding the gravity survey, we do not believe that the presented Bouguer anomalies allow only regional interpretations of the permeability structure of the basement. Instead, the lateral extent of anomalies is in the order of a few hundred meters, which corresponds to the scale of potential geothermal reservoirs of operation EGS plants.

*What geomechanically happen if the fault observed on the outcrop is buried several km deep? We know from reservoir studies that the permeability of the fault zone can be modified if they are critically stressed. Did past-circulations occurred in these faults? If yes, what is the nature of the*

*fluid? Are they opened, sealed or healed fault zone? These are questions that should at least be mentioned.*

Answer: Major faults are not exposed in the area, making it impossible for us to obtain reliable information on fluid circulation patterns and mineralization (see also comment below). As noted in the manuscript, stress magnitudes are also largely unconstrained in the study area, which means that we cannot assess with any certainty whether the faults are critically stressed. To overcome this lack of information, we conducted the DFN parameter study, which allows us to examine the range of hydraulic properties of a buried basement with the same fracture network properties as in the Tromm Granite. We have revised the discussion considerably to show this more clearly.

*They propose to apply their results for a future geothermal project in the Odenwald area and often mentioned the projects in operation of the URG as example. However, they forgot to really confront their results to observations made in the URG when they define the ideal targets for geothermal projects (comments 9, 14, 16 and 18). Moreover, when you conduct a geothermal prospection, you cannot ignore the thermal data of the area, that is not even mentioned in the study.*

Answer: We revised the manuscript accordingly. See specific comments below.

*I can only recommend this manuscript for publication after moderate revisions suggested below that the authors may want to consider when improving their paper.*

*Comment 1: L28*

*Exploration, therefore, began already in the 1980s, allowing to build on decades of experience (Reinecker et al., 2019; Cuenot et al., 2008; Dezayes et al., 2005a)*

*Please check also references Genter et al. 2010, C.R. Geosci. Doi:10.1016/j.crte.2010.01.006. & Schill et al., 2017, Geothermics, Doi: 10.1016/j.geothermics.2017.06.003.*

Answer: We added both references.

*Comment 2: L30*

*Convective heat transport along active large-scale fault zones has been identified as the main reason for the elevated temperatures at reservoir depths of 2 to 4 km (Bächler et al., 2003; Baillieux et al., 2013; Guillou-Frottier et al., 2013; Duwiquet et al., 2021)*

*Please rephrase this sentence. The thermal gradients below 1km depth is 5°C/km in Soultz and Rittershoffen wells and almost null below 2km in Bruchsal, Landau, Cronenbourg wells. Thus, between 2 and 4km depth, the temperature does not increase so much. The temperature increases anormally in the shallow part of the sedimentary cover.*

Answer: We rephrased this part.

*Comment 3: L45-50*

*The authors seem not to be aware that borehole imaging are rarely investigated alone. At the borehole scale, gamma ray and cuttings samples are almost always available in deep wells and they inform about the mineralogy and alteration petrography very precisely. These informations are correlated to*

*the structural ones from borehole imaging. Besides looking mineralization observed on an outcrop cannot be simply transposed to the reservoir. Temperature and pressure vary very quickly from a fracture to another one and influence the minerals deposition within. Please nuance your affirmations.*

Answer: We rephrased this part.

*Comment 4: L133 c. 20 × 30 m*

*I don't understand how a pixel cannot be a square. Why do you use 2 DEMs and not only the high resolution one?*

Answer: The SRTM model is provided in geographic coordinates with one arc second resolution. In this coordinate system the pixels are indeed square, but when converting the model to UTM coordinates, they become rectangular. We removed the specifications in meters to avoid misunderstandings.

We use two DEMs with varying resolution, because they allow the detection of different features. While rather local lineaments are analyzed with the 1m DEM, the SRTM models serves to analyses the regional lineaments. Several previous studies have adapted this approach and are cited in the text.

*Comment 5: paragraph 3.1.2*

*Where are the results of the automatic plane recognition? Why did you manually interpret the data besides the automatic interpretation? Do the results differ? Is Figure 3b the result of the calculation described L155? If yes please refer in the text. I find the methodology section for the LiDAR data very confusing and don't really understand for which purpose are made the different treatments of the data. Please clarify this section.*

Answer: these two approaches investigate different properties of the fracture network and are therefore not comparable. The automatic plane recognition is done to determine just the fracture plane orientation (Figure 5). The manual interpretation is done to study the length distribution, the fracture density/intensity and connectivity, as this cannot be generated automatically so far. Figure 3 is a combination of the automatic and manual interpretation. We slightly adjusted the text of this sub-chapter to make it clearer.

*Comment 6: is significantly highly than the amount of mapped faults*

*Do you mean significantly "higher" than?*

Answer: We corrected this part.

*Comment 7: 177 for SRTM and 471 for the 1m DEM*

*The 177 lineaments from SRTM are also observed and counted in the 471 with the 1m DEM?*

Answer: There is some overlap between the lineaments from the SRTM and the 1m DEM, but they are counted separately.

*Comment 8: figure 4 (b) regional analysis using SRTM data (20 x 30 m resolution) (adapted from van Zyl, 2001); (c) local analysis using 1 m DEM (adapted from HVBG, 2021).*

*From the paragraph 3.1.1, I understood that you inspected the lineaments with QGis. On figure 4b, you mention "adapted from van Zyl". A SRTM is a free data, so what did you adapt from this reference? I guess HVGB gave you the 1m DEM of 4c, but only the raw data or also an interpretation of the lineaments? What did you do really on the figure 4? Please clarify.*

Answer: We clarified this in the manuscript. The sources provided the raw DEM data.

*Comment 9: L340 the mean permeabilities of the granitic basement range from 1E-17 to 1E-15 m2 (Egert et al., 2020; Mahmoodpour et al., 2021 in prep.)*

*Please add the value for the permeability in the fault from Baujard, C. et al. 2017 (Geothermics, doi:10.1016/j. geothermics.2016.11.001).*

Answer: The reference was added.

*Comment 10: L371*

*Fig 10 is called before Fig 9*

Answer: We removed this sentence as it is repeated later in the text (section 4.2.3)

*Comment 11: L396-398*

*Do you have a reference from literature for this affirmation or it is something you observed on the field?*

Answer: We removed this paragraph from the main text, as it is too speculative.

*Comment 12: L418 Contrary to this, the fracture sets of the two outcrops Hammelbach and Weschnitz Valley are more influenced by local fault zones.*

*Do you have a reference for the direction of the local fault zones in the area or do you hypothesize that they are striking between N50°E and N130°E and thus, influence the sets of the two outcrops? Please clarify*

Answer: This was inferred from the lineament analysis, where features with this orientation can clearly be recognized. We adjusted the text accordingly.

*Comment 13: plannend*

*Correct with "planned"*

Answer: Was corrected

*Comment 14: L466 This observation suggests that the fault core does not show increased permeability compared to the damage damage zone (Caine et al., 1996), e.g. due to mineralization, which also explains the positive gravity anomalies.*

*Please delete one "damage" of your sentence.*

Answer: The sentence was largely rephrased.

*Correlation between acoustic logs and hydraulic tests in GPK-1 at Soultz reveals that the fluid flow is channelized in the core of the fracture zone. (Evans, K., Genter, A., Sausse, J., 2005. J. Geophys. Res. doi:10.1029/2004JB003168.) In the model of Caine et al. (1996), the permeability of the core is directly linked to the nature of the mineralization. Quartz is associated with a high permeability whereas, occurences of clays tend to reduce the permeability. Did you observe the mineralization of the studied fault zone?*

Answer: As no major fault is directly outcropping in the study area, we cannot make definitive statements about the nature of the mineralization. This is the reason why we choose indirect geophysical methods for the characterization. We state this now more clearly in the introduction.

*Comment 15: On the one hand, the granitic body has a similar mineralogical composition as the reservoir rocks e.g. in Soultz-sous-ForeÌ‚ts or Rittershoffen (Dezayes et al., 2005b).*

*Please for the mineralogical composition of Soultz, cite " Traineau, H., Genter, A., Cautru, J.-P., Fabriol, H., CheÌ€vremont, P., 1992. Petrography of the granite massif from drill cutting analysis and well log interpretation in the geothermal HDR borehole GPK-1 (Soultz, Alsace, France). In: Bresee, JamesC. (Ed.), Geothermal Energy in Europe – The Soultz Hot Dry Rock Project. Gordon and Breach Science Publishers, Montreux, Switzerland, pp. 1–29."*

*and Rittershoffen cite "Vidal et al., 2018, JVGR, doi : 10.1016/j.jvolgeores.2017.10.019"*

Answer: references were added.

*Comment 16: L475 Consequently, the orientation and density of fractures in the deep boreholes can assumed to be similar to the Odenwald values.*

*Which deep boreholes? The orientation E-W observed at the North of the Tromm Granite is not a predominant set of the deep wells in the URG.*

Answer: We largely revised chapter 5.3 and removed this sentence from the manuscript.

*Comment 17: L486 which fits well with obervations from Soultz sous Forets (Egert et al., 2020; Sausse et al., 2010)*

*Please correct "observations"*

Answer: Was corrected

*Please cite Evans, K., Genter, A., Sausse, J., 2005. J. Geophys. Res. doi:10.1029/2004JB003168. Inside the paper, it is described that in GPK-1, 95% of the flow entered the rock mass at 10 discrete flow points over more than 500 fractures detected in image logs.*

Answer: reference was added to the text.

*Comment 18 L488 The deviated open-hole sections should preferably be oriented perpendicular to the fracture set with the highest permeability, thus intersecting as many of them as possible, allowing for high flow rates.*

*This is not observed at Rittershoffen. The well GRT-2 presents the highest flowrate of the URG and the trajectory of the well is N-S inside the Rittershoffen fault zone that strikes N-S. The well and the fault are parallel, but the well was drilled inside the fault zone that maximize the connection between well and permeable fracture network associated to the fault. Please check the results of the references you cited "Baujard et al., 2017" and "Vidal et al., 2017" and nuance your affirmation in your manuscript.*

Answer: We largely revised chapter 5.3 including this part.

*Comment 19 L491 Controlling induced seismicity*

Please rephrase, because you cannot "control" seismicity, maybe you mean minimize?

Answer: We rephrased this part.

*Comment 20 L493 Stimulation is generally more feasible in areas with higher natural permeability*

*Please rephrase, because if natural permeability is high, you don't need to stimulate.*

Answer: We rephrased 'higher natural permeabilities' to 'naturally elevated permeabilities'. It makes little sense to stimulate a reservoir where the initial permeability is almost zero, as high injection pressures would be required that would lead to increased seismicity. The permeability should therefore be somewhat elevated compared to the average basement rocks already before stimulation.

*Comment 21 Geothermal prospection is also based on the temperature data. You can always try to enhance the hydraulic properties of your reservoir with EGS technology, but the temperature is something you cannot modify so you need to secure this parameter during your exploration study.*

Answer: We generally agree with this comment. However, the focus of this study is on the structural and potential hydrogeological properties of the basement. We did not apply methods to evaluate the temperature field in the specific area, but we are currently working on a quantitative techno-economic resource assessment for the crystalline basement in the northern URG, where the temperature is of course a major input factor. The results will be published in a subsequent paper. We added this aspect to the discussion.

*Comment 23 L528 Structural investigations and gravity anomalies show that the most suitable hydraulic properties are expected at the margin of the granitic pluton, where faults of regional extent influence the fracture network.*

*During the cooling of a granitic pluton, the margins are more affected by circulations of the residual fluids and thus, more altered than the center. The altered zones are preferential pathways for further circulations of the geothermal resource.*

Answer: We added this aspect to the manuscript.

*Comment 24 L530 Granite constitutes a relatively intact basement with significantly lower permeability, where permeability could only be created artificially by stimulation measures for geothermal utilization.*

*You cannot create permeability artificially with stimulation. Stimulation is for enhancing existing natural permeability. Fracturation can "create" the permeability but in the case of the URG, this technique is not appropriate because induced seismicity in urbanized areas generally lead to the end of the geothermal project.*

Answer: We removed this statement from the conclusions